# NRF2 in the Epidermal Pigmentary System

**DOI:** 10.3390/biom13010020

**Published:** 2022-12-22

**Authors:** Tatsuya Ogawa, Yosuke Ishitsuka

**Affiliations:** 1Department of Dermatology, Faculty of Medicine, University of Tsukuba, 1-1-1 Tennodai, Tsukuba 305-8575, Japan; 2Department of Dermatology, Osaka University Graduate School of Medicine, 2-2 Yamadaoka, Suita 565-0871, Japan

**Keywords:** antioxidant, melanocyte, NRF2, oxidative stress, reactive oxygen species, redox, vitiligo

## Abstract

Melanogenesis is a major part of the environmental responses and tissue development of the integumentary system. The balance between reduction and oxidation (redox) governs pigmentary responses, for which coordination among epidermal resident cells is indispensable. Here, we review the current understanding of melanocyte biology with a particular focus on the “master regulator” of oxidative stress responses (i.e., the Kelch-like erythroid cell-derived protein with cap‘n’collar homology-associated protein 1-nuclear factor erythroid-2-related factor 2 system) and the autoimmune pigment disorder vitiligo. Our investigation revealed that the former is essential in pigmentogenesis, whereas the latter results from unbalanced redox homeostasis and/or defective intercellular communication in the interfollicular epidermis (IFE). Finally, we propose a model in which keratinocytes provide a “niche” for differentiated melanocytes and may “imprint” IFE pigmentation.

## 1. Introduction

Oxidative stress results from exposure to reactive oxygen intermediates, such as superoxide anions (O_2_^−^), hydrogen peroxide (H_2_O_2_), and hydroxyl radicals (HO), and can damage proteins, nucleic acids, and cell membranes, leading to mutagenesis or cell death [1]. Reactive oxygen species (ROS) are produced by mitochondria and peroxisomes during cellular physiological metabolism. ROS-induced cumulative damage eventually causes numerous diseases. The skin serves as the interface between an organism and its external environment, protecting against xenobiotics or ultraviolet (UV) radiation, which can cause ROS-mediated tissue damage. Melanocytes produce pigmentation in hair and the interfollicular epidermis (IFE), thereby contributing to photoprotection and thermoregulation [2]. Vitiligo is an autoimmune-driven chronic depigmenting disease in which oxidative stress plays important roles [3,4]. The Kelch-like erythroid cell-derived protein with cap‘n’collar homology-associated protein 1 (KEAP1)-nuclear factor erythroid-2-related factor 2 (NRF2) system is a major antioxidative apparatus whose activation largely depends on the oxidation of cysteine residues on the actin-anchored KEAP1 protein [5]. NRF2 not only affects melanocyte proliferation/differentiation [6] but also helps melanocytes survive the cytotoxic immune responses in vitiligo [7]. Thus, because reduction and oxidation (redox) governs myriad biological processes, NRF2 appears to have broad and significant roles in melanocyte biology. Here, we sought to summarize the significant role of the KEAP1-NRF2 system, whose activation is tightly controlled by the redox status of thiol and disulfide.

## 2. The KEAP1-NRF2 System as a Master Regulator of Redox Homeostasis

Redox refers to the give-and-take of electrons between molecules (and/or their moieties). Oxidants deprive the target of electrons and are thus called electrophiles. Reductants provide the target with electrons and are therefore called electron donors. Therefore, “antioxidants” are substances that rescue organisms from excessive oxidation, directly or indirectly. The former, such as the glutathione (GSH) precursor *N*-acetyl cysteine, replenishes the intracellular GSH pool, and the latter is often electrophilic and strongly induces cellular antioxidative defenses (such as sulforaphane), resulting in the production of antioxidative effectors (e.g., glutamate-cysteine ligase catalytic subunit [GCLC]) from cells in stress.

KEAP1 is an actin-anchored cysteine-rich protein that senses extra- or intracellular electrophilic milieus causing “oxidative damage,” and NRF2 is a ubiquitous transcription factor [5]. In the inactivated state, NRF2 is polyubiquitinated and degraded by proteasomes. During periods of stress, NRF2 exerts counter-responses. For instance, NRF2 transactivates cystine/glutamate transporter (solute carrier family 7 member 11 [*SLC7A11*]) [8] and upregulates GSH-synthesizing enzymes (*GCLC* and glutamate-cysteine ligase modifier subunit [*GCLM*]) and replenishes intracellular GSH storage [5]. Since the identification of the *cis*-acting antioxidant (electrophile) response element, many genes have been identified as downstream effectors of the KEAP1-NRF2 system, including ROS quenchers (thioredoxin reductase 1 [*TXNRD1*] and peroxiredoxin 1 [*PRDX1*]), phase II detoxification mediators (GSH S-transferase [*GST*], NAD(P)H quinone dehydrogenase 1 [*NQO1*], and heme oxygenase-1 [*HMOX1*]), transmembrane drug transporters (multidrug resistance-associated protein 1 [*MRP1*]), and structural proteins related to keratinization (small proline-rich proteins [*SPRR*s] and late cornified envelopes [*LCEs*]) [5,9].

## 3. Melanocyte Biology

Melanocytes originate from the neural crest and localize in the hair follicles (HFs) and IFE, endowing the body surface with pigmentation [2]. Human melanocytes exist in the IFE and HFs, whereas murine melanocytes exclusively reside in pelage HFs (exceptions are the ears, nose, paws, tail, etc.). Approximately 1200 melanocytes reside per square millimeter of human skin, regardless of race [10]. Epidermal melanocytes exist in the basal IFE layer and form the epidermal melanin unit, in which one melanocyte communicates with 30–40 IFE keratinocytes (KCs) [10]. Melanocytes adhere to KCs via adhesion molecules such as E-cadherin (Ecad) and desmoglein 1 (DSG1) [11]. There are two distinct melanocytic populations in HFs: melanocyte stem cells (McSCs) and their differentiated progeny. McSCs reside in the hair bulge and secondary hair germ (the lower permanent portion of the HFs) and show cyclic activity in parallel with HF stem cells [12].

The crosstalk between melanocytes and KCs through secreted factors (e.g., stem cell factor [SCF], basic fibroblast growth factor [bFGF], granulocyte macrophage-colony stimulating factor, endothelin 1, α-melanocyte-stimulating hormone [α-MSH], prostaglandin E_2_, prostaglandin F_2α_, and nerve growth factor) and cell–cell interaction controls melanocyte proliferation, differentiation, melanogenesis, and dendritogenesis [10,13,14]. Melanocytes are also controlled by dermal fibroblasts via secreted factors (e.g., SCF, bFGF, and hepatocyte growth factor) [10,13,14].

Melanocytes exhibit immunological characteristics. They can express major histocompatibility complex classes I and II, immunomodulatory adhesion molecules (e.g., intercellular adhesion molecule-1 and vascular cell adhesion molecule-1), and the costimulatory receptor CD40 [15,16,17,18]. In addition, melanocytes can produce several cytokines such as interleukin (IL)-1, IL-6, IL-8, and transforming growth factor-β1 [19,20,21].

### 3.1. Melanogenesis

Melanogenesis (melanin synthesis) is a complex biosynthetic process [22] that occurs predominantly in a lysosome-like organelle called the melanosome [23] (Figure 1). Melanosomes are exported from melanocytes to adjacent KCs, and pigmentation differences occur owing to variations in the size, composition, and distribution of melanosomes [2]. Melanin has many biological functions, including UV light absorption/scattering, free radical scavenging, coupled redox reactions, and ion storage [10]. Melanin comes in two forms: yellow to red pheomelanin and brown to black eumelanin. The metabolic cascade of melanogenesis begins with the hydroxylation of the amino acid tyrosine to L-3,4-dihydroxyphenylalanine, which is then converted into dopaquinone by tyrosinase (TYR) [10]. This series of reactions involve physical contact between the melanosome and mitochondria [24], inherently generating ROS, such as O^2−^ or H_2_O_2_ [25]. Dopaquinone oxidation produces dopachrome, which is decarboxylated into 5,6-dihydroxyindole or tautomerized to 5,6-dihydroxyindole-2-carboxylic acid by tyrosinase-related protein 2 (dopachrome tautomerase or TYRP2). TYR and tyrosinase-related protein 1 (TYRP1) catalyze further reactions to synthesize eumelanin [26]. The relative abundance of GSH/cysteine promotes the conversion of dopaquinone into cysteinyldopa, preferentially generating pheomelanin. Conversely, the relative abundance of glutathione disulfide (GSSG)/cystine (two cysteines bridged via disulfide) promotes melanosome maturation, yielding stable, polymerized eumelanin. In summary, like other biological processes, the redox milieu appears to play central roles in melanogenesis and thus skin pigmentation/depigmentation.

### 3.2. Redox “Switch” in Skin Pigmentation

In general, the pheomelanin synthetic pathway requires cysteine/GSH, whereas the eumelanin pathway is expedited in the absence of thiols (high cystine/cysteine or GSSG/GSH ratios) [27,28]. Compared with pheomelanin, eumelanin is more polymerized and structurally stable, and is better at protecting against ROS. Thus, darkly pigmented eumelanin serves as a superior antioxidant/photoprotector on the skin surface [10]. Melanogenesis consumes high energy, produces O^2−^ or H_2_O_2_ from mitochondria [24,25], and generates a pro-oxidant state, potentially sensitizing the epidermis to oxidative stress [22] (Figure 1). An unbalanced extracellular redox milieu can also perturb cellular fate decisions, leading to cell death (apoptosis) [29] or proliferation (melanomagenesis) [30]. A rather perplexing fact is that pheomelanin has a potent photosensitizing capacity [31]. Pheomelanin toxicity likely causes premature aging and cutaneous tumorigenesis in fair-skinned individuals. In contrast, eumelanin can enhance permeability barrier function by lowering the surface pH of darkly pigmented human [32]/mouse [33] skin.

A proper antioxidative response induced by changes in the intra-/extracellular redox milieu appears to give an important direction regarding the trajectory of the pigmentary pathways. Defective transmembrane cystine transport caused by the subtle gray (*sut*) mutation in the *Slc7a11* gene reduces intracellular GSH levels and increases GSSG levels (thus, the GSH/GSSG ratio decreases). The failure in response against extracellular oxidative signals skews melanogenesis toward the eumelanin synthesis pathway [34]. The mitochondrial redox-regulating enzyme nicotinamide nucleotide transhydrogenase (*NNT*) controls melanosomal maturation (and eumelanogenesis) [35]. This inner mitochondrial membrane protein regulates NAD(P)^+^/NAD(P)H homeostasis by mediating electron transfer [36]. Although NNT can bring about an intracellular oxidative milieu in certain circumstances [37], NNT increases GSH/GSSG ratios and negatively regulates eumelanogenesis [35], similar to what has been observed for NRF2 [6]/*Slc7a11* (*sut* mice) [8]. However, the most striking facts are that single nucleotide polymorphisms (SNPs) in the *NNT* gene clearly differentiate skin pigmentary phenotypes, and inhibition takes place at the post-transcriptional levels [35]. Existing evidence argues against the classic pigmentation pathway dependent on the UV-cyclic adenosine monophosphate-microphthalmia-associated transcription factor (MITF) axis, which in turn transactivates *TYRP1* and *TYRP2* [35]. We should note that *sut* melanocytes, which mount suboptimal counter-responses against an extracellular oxidative milieu [8], also exhibit abnormal proliferation and differentiation in vitro [34]. Thus, redox milieus (intracellular or extracellular) can profoundly affect melanocyte biological behaviors and fate decisions [38]. In summary, investigating the redox “switch” in skin pigmentary pathways not only could lead to a profound understanding of skin pigmentation biology but also may pave a way toward repurposing small molecule inhibitors for diverse pigmentary disorders [35].

## 4. Vitiligo

Vitiligo is an acquired chronic pigmentary disorder that affects 0.5% to 2% of the world’s population without a clear preference for race or sex [39]. The US population-based prevalence estimate of vitiligo in adults was between 0.76% and 1.11% [40]. Vitiligo results from selective melanocyte loss, which leads to pigment dilution in the affected skin and mucosa. Typical vitiligo lesions present as milky-white nonscaly macules with distinct margins. Generally, vitiligo is clinically diagnosed, and no laboratory tests or biopsies are required. Two major forms of the disease are well-recognized according to the distribution of lesions: segmental vitiligo (SV) and non-segmental vitiligo (NSV) [41]. NSV includes acrofacial, mucosal, generalized, universal, mixed, and rare variants. Distinguishing SV from other types of vitiligo is important because of its prognostic implications [42].

Vitiligo pathogenesis involves multiple factors, such as genetic background, metabolic abnormalities, oxidative stress, generation of inflammatory mediators, autoimmune responses, and decreased melanocyte adhesiveness [43]. These multiple mechanisms may function collectively, leading to melanocyte destruction [42].

### 4.1. Genetic Background of Vitiligo

Multiple studies have revealed the genetic background of vitiligo development. Approximately 50 different genetic loci that contribute to the risk of vitiligo have been discovered, principally in European-derived whites and Chinese [44]. A genome-wide association study identified the susceptibility loci for autoimmunity (e.g., *HLA* classes 1 and 2, *PTPN22*, *IL2R* α, *GZMB*, *FOXP3*, *BACH2*, *CD80*, and *CCR6*) and melanocyte-specific gene *TYR* in patients with vitiligo [45]. In addition, altered *NALP1* (the gene encoding NACHT leucine-rich repeat protein 1), a regulator of innate immunity, was found to be a risk factor for vitiligo [46]. Recently, polymorphic expression of *MTHFR* (the gene encoding methylene tetrahydrofolate reductase), which regulates homocysteine levels, has been identified in patients with vitiligo [47]. *XBP1P1* (the gene encoding X-box binding protein 1) has also been associated with vitiligo. It is pivotal in attenuating the unfolded protein response and driving stress-induced inflammation in vivo [48].

### 4.2. Oxidative Stress in Vitiligo

In vivo and in vitro investigations have confirmed widespread alterations of the antioxidant system in the skin and blood of patients with vitiligo [7,49,50,51,52,53,54,55,56,57,58,59,60,61,62,63], suggesting important roles of oxidative stress in vitiligo onset and progression [4,64]. Melanocytic oxidative stress can cause a local inflammatory response and activate innate immunity through damage-associated molecular patterns (DAMPs), generating melanocyte-specific cytotoxic immune responses in genetically predisposed individuals [43]. Vitiligo melanocytes have decreased Ecad and increased expression of the anti-adhesion molecule tenascin [65,66] and are susceptible to oxidative stress [55] or UVB [67]. Given that melanocyte-KC interaction is vital for skin pigmentation, oxidative stress-driven melanocyte dysadhesion may represent an initial pathogenic event, which further precipitates oxidative damage [55,67,68] and leads to senescence (degeneration) [43]. Alterations in TYRP1 synthesis/processing impair eumelanogenesis and hamper melanosome maturation, increasing melanocyte oxidative damage [67]. Although mitochondria-derived ROS mediates aging or apoptosis in healthy cells [43], it is the culprit for vitiligo melanocyte dysfunction [69,70]. Oxidative stress impairs the function of membrane lipids and cellular proteins in melanocytes [67,68]. Biopterin synthesis and recycling are altered (i.e., increased production of 6-tetrahydrobiopterin and 7-tetrahydrobiopterin), leading to further oxidative stress and cell damage [71]. Oxidative stress can also affect McSCs in HFs, which might lead to a higher incidence of early hair graying in patients with vitiligo [72].

Recently, a molecular mechanism involved in oxidative stress-induced melanocyte degeneration has been proposed. Oxeiptosis is an apoptosis-like, nonclassical, ROS-induced cell death pathway [73]. Because H_2_O_2_ induces vitiligo melanocyte cell death, oxeiptosis may contribute to vitiligo pathogenesis [74]. The microRNA (miRNA) miR-25 suppresses MITF levels in melanocytes and SCF and bFGF expression in KCs, thereby contributing to melanocyte degeneration [75].

### 4.3. Immune Activation in Vitiligo

Autoimmunity has been implicated in vitiligo pathogenesis [76]. This is supported by the presence of antibodies against melanocytes, association with polymorphism at immune loci, prominent T cell infiltration in perilesional areas, cytokine expression, and association with other autoimmune diseases (e.g., autoimmune thyroiditis and type 1 diabetes mellitus) [43].

Oxidative stress connects innate immunity and adaptive immunity [77,78]. The activation of innate immune cells occurs upon sensing exogenously or endogenous stress signals, mainly from melanocytes [43]. ROS overproduction causes melanocytes to release exosomes [79] containing melanocyte-specific antigens, miRNAs, heat shock proteins, and other proteins that act as DAMPs [42]. These exosomes activate nearby dendritic cells and induce their maturation [80,81,82], followed by cytokine- and chemokine-driven T helper 17 cell activation [83,84,85] and regulatory T cell dysfunction [86,87]. Cytotoxic CD8^+^ T cells that target melanocytes are responsible for melanocyte destruction [88]. CD8^+^ T cells produce several cytokines, including interferon (IFN)-γ [88,89]. IFN-γ signaling activates the Janus kinase (JAK)-signal transducer and activator of transcription (STAT) pathway, leading to CXC chemokine ligand 9 (CXCL9) and CXCL10 secretion in the skin [90]. CXCL9 promotes the bulk recruitment of melanocyte-specific CD8^+^ cytotoxic T cells (CTLs) to the skin [90,91], whereas CXCL10 promotes the effector function of CD8^+^ T cells and their localization within the epidermis [90,91].

### 4.4. NRF2 in Vitiligo Pathology

NRF2 protects against cellular oxidative damage [5]. This principle makes the idea of supporting vitiligo melanocyte survival legitimate. NRF2 overexpression via transfection of NRF2 gene-containing plasmids (pCMV6-XL5-Nrf2) protects the immortalized human melanocyte cell line PIG1 against H_2_O_2_-induced oxidative stress [92]. α-MSH induces NRF2 and protects normal human epidermal melanocytes (NHEMs) against UVB-induced oxidative stress [93]. NRF2 activation by small interfering RNA-mediated KEAP1 knockdown protects NHEMs against toxicity induced by the depigmenting agent monobenzone (monobenzyl ether of hydroquinone [MBEH]) [94]. Augmented NRF2 signaling could replenish the cellular GSH pool (thus increasing the GSH/GSSG ratio) [5]. Accordingly, adenovirus-mediated NRF2 overexpression inhibits melanogenesis in NHEMs [6]. Conversely, knockdown of *NRF2* or its downstream genes, such as *NQO1* and *PRDX6* reduces cultured melanocyte viability [94]. Melanocytes isolated from *sut* mice that harbor the mutation in an NRF2-target *Slc7a11* gene [8] exhibit reduced rates of melanocyte viability/proliferation [34]. Moreover, *sut* mice exhibit reduced hair pheomelanin content, whereas that of eumelanin is barely affected although its effect on IFE pigmentation (tanning response) has not been assessed [34]. Furthermore, H_2_O_2_-induced oxidative stress promotes cytoplasmic translocation/release of high mobility group box 1 from NHEMs, resulting in indirect activation of NRF2 and its target genes (*HMOX1*, *NQO1*, *GCLC*, and *GCLM*) [95].

Evidence indicates that antioxidative response aberrations are central to vitiligo pathogenesis. SNPs in the NRF2 promoter may increase vitiligo risk [96], suggesting that aberrant antioxidative responses can be genetically determined. The epidermis of patients with vitiligo harbors increased H_2_O_2_ levels [58,61], and the lesional epidermis exhibits higher *NRF2*, *NQO1*, *GCLC*, and *GCLM* expression levels compared with the non-lesional epidermis [97]. An enhanced oxidative damage may increase the vulnerability of melanocytes to oxidative damage [7,94], which can be counteracted by local HO-1 augmentation using psoralen plus UVA (PUVA) treatment [98]. However, IL-2-induced expansion of circulating melanocyte-specific CD8^+^ cytotoxic T cells (CTLs) [99] may bring about a reductive environment (thus reduces GSH/GSSG ratios) [100], decreasing serum HO-1 levels [7]. Collectively, the vitiligo lesional epidermis appears to suffer from high oxidative damage, which may in turn dampen normal antioxidative responses. This notion is further supported by the attenuated induction of phase II detoxification genes in the lesional skin after in vitro and ex vivo treatment with the electrophilic compounds curcumin and santalol [97]. In this meticulous experimental setting, isolated KCs were found to be more susceptible to apoptosis, whereas melanocytes were relatively resistant against apoptosis [97]. These results suggest that melanocyte-KC communication sustains overall redox balance in the epidermis [38] as well as circulation [7].

Augmenting the antioxidative responses of melanocytes and supporting their survival by pharmacological means could be attractive measures. Several compounds can protect melanocytes against oxidative stress through NRF2 activation (Table 1). Melatonin and its metabolites (6-hydroxymelatonin, N1-acetyl-N2-formyl-5-methoxykynuramine, *N*-acetylserotonin, and 5-methoxytryptamine) [101], vitamins (folic acid [102], methylcobalamin [103], and vitamin D [104]), natural compounds (4-octyl itaconate [105], ginsenoside Rk1 [106], *Cistanche deserticola* polysaccharides [107], glycyrrhizin [108], *Lycium barbarum* polysaccharides [109], paeoniflorin [110], 6-shogaol [111], paeonol [112], afzelin [113], apigenin [114], baicalein [115], vitexin [116], and berberine [117]), and therapeutic agents (aspirin [118], dimethyl fumarate [DMF] [94], simvastatin [29], molecular hydrogen [52], and cold atmospheric plasma [119]) are shown to protect melanocytes against oxidative damage caused by H_2_O_2_, UVB, or MBEH. DMF is the methyl ester of fumaric acid and is one of the most successful NRF2 activators. DMF has been approved by the United States Food and Drug Administration (FDA) for relapsing-remitting multiple sclerosis, a demyelinating autoimmune disease [120]. The European Medicines Agency has also approved DMF to treat moderate-to-severe plaque psoriasis [121]. DMF treatment enhances NRF2 nuclear localization and protects NHEMs and vitiligo melanocytes against MBEH-induced oxidative stress [94]. However, topical application of DMF can cause contact dermatitis [122]. This might be attributable to NRF2-enhanced skin sensitization [77,78]. Therefore, caution should be taken when applying potent electrophilic chemicals percutaneously, and refined drug delivery systems may be needed.

### 4.5. Management of Vitiligo

Vitiligo is not “just a cosmetic condition” but is psychologically devastating and stigmatizing [123]. The psychological impact on quality of life (QOL) is similar to that of other skin diseases, such as atopic dermatitis and psoriasis [124].

The aim of clinical management is to halt the autoimmune-driven depigmentation and restore the homeostatic pigmentation. Treatment options depend on several factors, such as disease subtype, extent, distribution, activity, patient age, phototype, effect on QOL, and motivation for treatment [42]. These treatments include topical therapies (e.g., corticosteroids and calcineurin inhibitors), phototherapies (e.g., photochemotherapies, narrowband UVB [NB-UVB], and excimer lasers or lamps], oral therapies (e.g., steroids and other immunosuppressants), surgery, and combination therapies [125]. Treatments are graded from first- to fourth-line options [125]. First-line treatment consists of topical therapy with corticosteroids and calcineurin inhibitors; second-line treatment, NB-UVB, PUVA, and systemic steroid therapy; third-line treatment, surgical grafting techniques; and fourth-line treatment, depigmentation therapies.

Although the abovementioned roles of oxidative stress in vitiligo rationalize antioxidant-based treatment, evidence of the efficacy of this treatment is quite limited [125]. To achieve repigmentation, pseudocatalase, vitamin E, vitamin C, ubiquinone, lipoic acid, *Polypodium leucotomos*, catalase/superoxide dismutase combination, and *Ginkgo biloba* may be administered with or without UV therapy [125]. Since the discovery of the role of the IFN-γ signaling axis, several clinical trials involving JAK inhibitors have been conducted [126]. JAK inhibitors, which target the type II IFN signaling pathway, have been shown to stimulate repigmentation in patients with vitiligo [127,128,129]. Tofacitinib, ruxolitinib, and baricitinib are the three major JAK inhibitors used for vitiligo. Ruxolitinib, an inhibitor of Janus kinase 1 (JAK1) and 2 (JAK2), was recently approved by the FDA to treat NSV in adult and pediatric patients aged ≥ 12 years. Ruxolitinib cream resulted in repigmentation through 52 weeks in phase 2 [130] and 3 [131] trials; however, its use is accompanied by acne and pruritus at the application site. Large-scale, long-term studies are required to elucidate the effects and risks of ruxolitinib cream application for vitiligo treatment.

### 4.6. Conclusions

Recent progress in vitiligo research has paved the way for disease pathway-based therapy. The IFN-γ-JAK-STAT pathway drives vitiligo pathogenesis, and JAK inhibitors, which presumably inhibit the effector function of CD49a^+^ cytotoxic epidermal resident CD8^+^ T cells efficiently [132], hold promise for better management of this emotionally devastating ailment. In this review, we initially aimed to examine the role of the KEAP1-NRF2 system in melanocyte biology/vitiligo pathogenesis. It has turned out, however, that this role [5] appears too far-reaching to be a disease-specific pathway. Systemic activation of the KEAP1-NRF2 system by the *Keap1*-null mutation not only augmented phase II detoxification but also led to uncontrolled keratinization of the squamous epithelium (SE) [9]. We and others have characterized the roles of the KEAP1-NRF2 system in skin diseases involving aberrations in inflammation/keratinization (reviewed in [9]). The aggregated evidence underscores the prominent roles of the KEAP1-NRF2 system in epidermal biology. In summary, the NRF2/KEAP1 system is important in vitiligo but far more specific than a therapeutic target.

## 5. Future Directions

Cutaneous pigmentary/depigmentary disorders, such as vitiligo or lentigines, do not necessarily accompany aberrant keratinization or acanthosis. Nonetheless, when considering the SE as a pigmentary “unit” [10], similar to classic immune cell components (the epidermal proliferation [differentiation] unit comprising KCs and Langerhans cells) [133], the epidermal “niche” being the ultimate determinant of cellular behavior may be evident [38]. Previous reports support this notion; unlike other minor epidermal residents, melanocytes express the desmosomal cadherin DSG1 [134], one of the critical commitment factors of IFE differentiation [135]. Loss of DSG1 in epidermal KCs may lead to melanocyte loss from the epithelium and promote invasive/metastatic growth of transformed melanocytes (melanoma) [134,136], suggesting that IFE KCs “imprint” (or instruct) melanocytic behaviors. This reasoning is further supported by the classic morphological changes in epidermal melanosomes following the topical application of the antipolymerization agent 4-tertiary butyl catechol [137], which could also augment the response of epidermal KCs to cellular distress (i.e., keratinization) [9]. This treatment blocks the eumelanin synthesis pathway, causing the appearance of immature pheomelanosomes in hairless mice [137]. Compared with pigmentation in the hair, tanning responses (IFE melanogenesis) depend on the nature of differentiated McSC-derived melanocytes; wet-surfaced SE (squamous mucosa), palmoplantar epidermis, or its apparatus (the nail) hardly experiences tanning responses (caused by the predominance of eumelanogenesis over pheomelanogenesis). The “niche” instruction or “structural imprinting” [138] aspect of pigmentogenesis would be further rationalized when IFE differentiation is analogized to sulfur metabolism; thiol groups of the proliferative layer are converted to disulfide polymerized keratins [139] (Figure 2). The principle, along with the possibility that the eumelanin synthetic pathway is regulated post-transcriptionally [35], tempted us to determine the “niche factor” within the IFE component. We have recently found that the IFE differentiating factor loricrin (LOR), which has a potent disulfide-linking capacity (reviewed in [138]), is indispensable for protection against UV radiation or electrophilic carcinogens (reviewed in [138]). Thus, we hypothesize that LOR could imprint the behaviors of IFE melanocytes and resident leukocytes [138]. The major effector of cornification (LOR) may act as a fate determinant of IFE melanocytes. Although further investigation and validation are required, revealing the hitherto unproven aspects of epidermal cell biology may lead to the development of mechanism-based skin pigmenting/depigmenting measures.

## Figures and Tables

**Figure 1 biomolecules-13-00020-f001:**
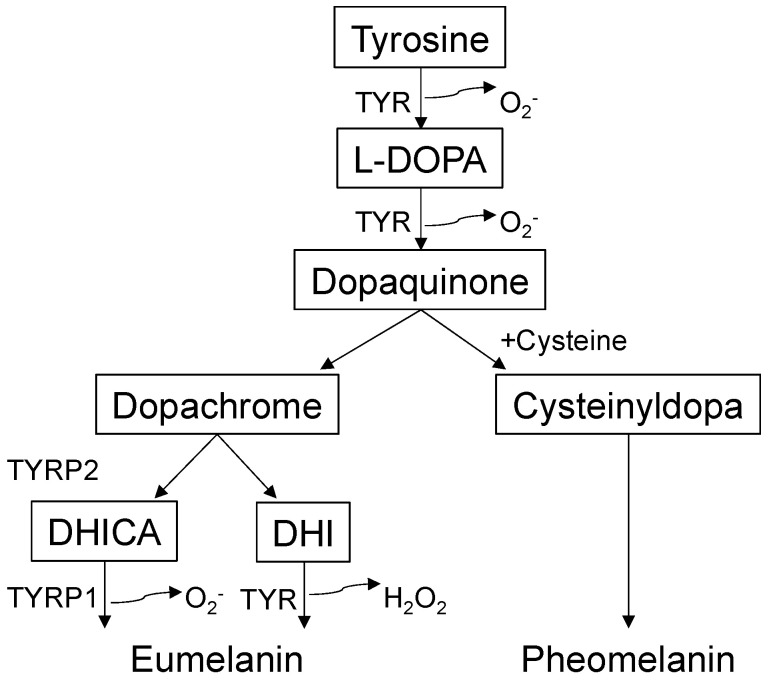
Overview of melanogenesis pathways. Tyrosinase (TYR), the rate-limiting enzyme for melanogenesis, oxidizes tyrosine to L-3,4-dihydroxyphenylalanine (L-DOPA) and dopaquinone. Dopaquinone reacts with excess cysteine/glutathione and generates cysteinyldopa, giving rise to pheomelanin. With cystine/glutathione disulfide abundance, dopaquinone generates dopachrome. Dopachrome undergoes spontaneous decarboxylation and forms 5,6-dihydroxyindole (DHI). In the presence of tyrosinase-related protein 2 (TYRP2), dopachrome produces 5,6-dihydroxyindole-2-carboxylic acid (DHICA) through tautomerization. TYR and tyrosinase-related protein 1 (TYRP1) catalyze further conversions, yielding eumelanin. The catalytic activity of TYR or TYRP1 generates superoxide anions (O_2_^−^) and hydrogen peroxide (H_2_O_2_).

**Figure 2 biomolecules-13-00020-f002:**
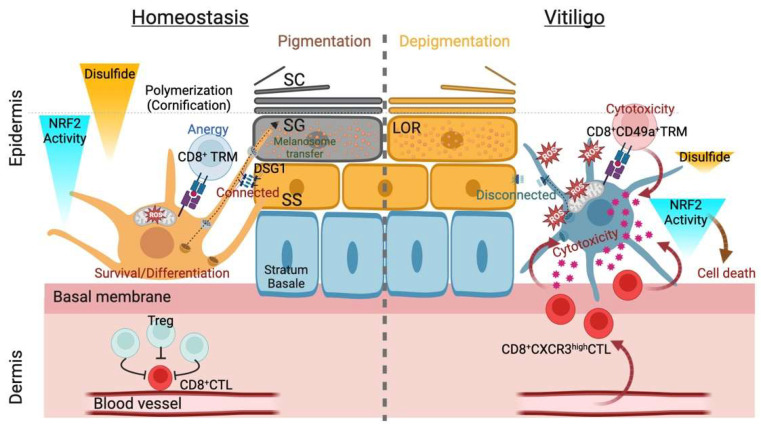
Summary of thiols/disulfides in the epidermal pigmentary system. Interfollicular epidermis (IFE) pigmentation largely depends on the balance between reduction and oxidation (redox) status with internal or external causes. The former refers to mitochondria-derived reactive oxygen species (ROS) during melanogenesis, and the latter may correspond to epidermal differentiation in which keratinocyte (KC) structural proteins undergo extensive disulfide bridge formation upon the initiation of cornification (transition from the stratum granulosum [SG] to the stratum corneum [SC]). Successful cornification largely depends on the biochemical nature of the structural protein loricrin (LOR, indicated as granules). Differentiating layer (stratum spinosum [SS] and SG)-specific desmosomal cadherin desmoglein 1 (DSG1) appears indispensable for functional IFE pigmentogenesis: melanosome transfer and maturation (eumelanogenesis) (**left**). In vitiligo melanocytes, stress from environmental factors (e.g., toxic chemicals) can cause aberrations in the IFE antioxidant systems in genetically predisposed individuals. Breached immune tolerance recruits melanocyte antigen-specific CD8^+^ cytotoxic T cells (CTLs) expressing CXC receptor type 3 (CXCR3) or CD49^+^ resident memory T cells (TRMs) in vitiligo (**right**), whereas the autoreactive CTLs constitutively become anergic with the help of regulatory T cells (Tregs) in homeostasis (**left**). The homeostatic gradient of nuclear factor erythroid-2-related factor 2 (NRF2)-mediated epidermal antioxidative defense and ensuing cornification yield a polymerized/pigmented SC, protecting against oxidative damage. However, local clonal expansion of CTLs in the IFE eventually eliminates melanocytes from the IFE niche (the epidermal melanin unit), perturbing the xenobiotic metabolism coordinated by NRF2 and resulting in depigmentation (leukoderma).

**Table 1 biomolecules-13-00020-t001:** NRF2-targeted therapy in vitiligo.

Treatment	Administration Route	Model	NRF2 Status	Effect	Reference
Hormone					
Melatonin	In vitro	UVB-treated NHEMs	Activation	Increases DNA repair and levels of p53 phosphorylated at serine 15 Reduces ROS generation	[101]
Vitamins					
Folic acid	In vitro	H_2_O_2_-treated PIG1	Activation	Increases cell viability and proliferation Decreases apoptosis and ROS generation Inhibits HMGB1	[102]
Methylcobalamin	In vitro	H_2_O_2_-treated PIG1	Activation	Increases cell viability and melanogenesis Decreases apoptosis and ROS generation	[103]
Vitamin D	In vitro	H_2_O_2_-treated PIG1 H_2_O_2_-treated PIG3V	Activation	Increases cell viability, proliferation, and migration Decreases apoptosis and ROS generation Activates Wnt/β-catenin signaling	[104]
Organic compounds					
4-Octyl itaconate	In vitro Intravenous	UVB-treated PIG1 UVB-treated HaCaT UVB-treated mice	Activation	Increases cell viability Decreases apoptosis and ROS generation Attenuates UVB-induced skin damage	[105]
Ginsenoside Rk1	In vitro	H_2_O_2_-treated PIG1	Activation	Increases cell viability Decreases apoptosis	[106]
Glycosides					
*Cistanche deserticola* polysaccharides	In vitro	H_2_O_2_-treated NHEMs H_2_O_2_-treated mouse melanoma B16F10 cells	Activation	Increase cell viability and melanogenesis Decrease cytotoxicity, apoptosis, and ROS generation	[107]
Glycyrrhizin	In vitro	H_2_O_2_-treated NHEMs	Activation	Increases cell viability Decreases apoptosis and ROS generation	[108]
*Lycium barbarum* polysaccharides	In vitro	H_2_O_2_-treated PIG1	Activation	Increase cell proliferation and melanogenesis Decrease apoptosis Activate the NRF2/p62 signaling pathway and induce autophagy	[109]
Paeoniflorin	In vitro	H_2_O_2_-treated PIG1 H_2_O_2_-treated PIG3V	Activation	Increases cell viability Decreases apoptosis and ROS generation	[110]
Polyphenols					
6-Shogaol	In vitro	H_2_O_2_-treated NHEMs	Activation	Increases cell viability and melanogenesis Inhibits apoptosis	[111]
Paeonol	In vitro	H_2_O_2_-treated PIG1	Activation	Increases cell viability and melanogenesis Decreases ROS generation and lipid peroxidation	[112]
Flavonoids					
Afzelin	In vitro	H_2_O_2_-treated NHEMs	Activation	Increases cell proliferation and phosphorylation of GSK-3β Decreases apoptosis, ROS generation, and lipid peroxidation	[113]
Apigenin	In vitro	H_2_O_2_-treated PIG3V	Activation	Increases cell viability Decreases lipid peroxidation	[114]
Baicalein	In vitro	H_2_O_2_-treated PIG3V	Activation	Increases cell viability Decreases apoptosis and mitochondrial dysfunction	[115]
Vitexin	In vitro	H_2_O_2_-treated PIG1	Activation	Increases cell viability and proliferation Decreases apoptosis, ROS generation, and IL-1β and IL-17A expression	[116]
Alkaloids					
Berberine	In vitro	H_2_O_2_-treated PIG1	Activation	Increases cell viability and melanogenesis Decreases apoptosis, ROS generation, and NF-κB activation	[117]
Therapeutic agents					
Aspirin	In vitro	H_2_O_2_-treated NHEMs	Activation	Increases cell viability Decreases apoptosis, ROS generation, and LDH release	[118]
Dimethyl fumarate	In vitro	MBEH-treated NHEMs MBEH-treated NLVMs MBEH-treated PLVMs	Activation	Increases cell viability	[94]
Simvastatin	In vitro	H_2_O_2_-treated NHEMs	Activation	Increases cell viability Decreases apoptosis and ROS generation Activates the MAPK pathway and p62	[29]
Molecular hydrogen	In vitro	Vitiligo epidermal cells H_2_O_2_-treated PIG1 H_2_O_2_-treated PIG3V H_2_O_2_-treated HaCaT	Activation	Increases cell viability, migration, melanogenesis, and mitochondrial function Decreases apoptosis, ROS generation, and lipid peroxidation	[52]
Cold atmospheric plasma	Topical	Vitiligo-like mouse model Vitiligo skin	Activation	Ameliorates vitiligo lesions in mice and patients Restores melanin distribution in mice and skin pigmentation in patients with vitiligo Decreases CD11c^+^ DC/CD3^+^ T cell/CD8^+^ T cell infiltration, CXCL10/IFN-γ/HIF-1α release, and inducible nitric oxide synthase in mice Increases gp100^+^ cells and decreases CD8^+^ T cells in patients	[119]

DC, dendritic cell; gp100, glycoprotein 100; GSK-3β, glycogen synthase kinase-3β; HIF-1α, hypoxia-inducible factor-1α; HMGB1, high mobility group box 1; H_2_O_2_, hydrogen peroxide; IL-1β, interleukin-1 β; IL-17A, interleukin-17A; LDH, lactate dehydrogenase; MAPK, mitogen-activated protein kinase; MBEH, monobenzyl ether of hydroquinone; NF-κB, nuclear factor-κB; NHEMs, normal human epidermal melanocytes; NLVMs, non-lesional vitiligo melanocytes; PLVMs, perilesional vitiligo melanocytes; UVB, ultraviolet B.

## Data Availability

Not applicable.

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
