# Peer review of "NRF2 in the Epidermal Pigmentary System"

_biomolecules, 2022, doi:10.3390/biom13010020_

Round 1

Reviewer 1 Report

The manuscript “NRF2 in Melanocyte Biology and Vitiligo Pathogenesis” describes the involvement of the NRF2/KEAP1 system in the pathogenesis of vitiligo and possible therapeutic approaches related to NRF2. The authors have briefly overviewed the NRF2/KEAP1 system, the melanocytes biology, and in vitiligo: the genetic predisposition, the involvement of oxidative stress, NRF2, and immunity. The paper is interesting, dealing with an important issue, but certain points could be improved.

Specific comments:

1. Section 2. “The KEAP1-NRF2 System for Maintaining Redox Homeostasis” could be more clearly written. The authors state: “The cytoplasmic protein KEAP1 senses oxidative insults, whereas the transcription factor NRF2 exerts counter responses [4]. In the steady state, NRF2 is polyubiquitinated and degraded by proteasomes. Upon sensing electrophiles or ROS, the reactive cysteine residues of KEAP1 are covalently modified, resulting in NRF2 stabilization….”

 What do the authors mean by KEAP1 senses oxidative insults while NRF2 exerts counter-responses?

In addition, when describing the NRF2/KEAP1 system in a steady state, how are covalent modifications of KEAP1 cysteine residues stabilizing NRF2? Which NRF2 activates NRF2 target genes?

2. Page 3, lines 139-140, sentence “ROS are released from melanocytes in response to stress and can lead to impaired expression or activity of the antioxidant system.” Unclear sentence. How is ROS released from melanocytes in response to stress?

3. The authors should discuss in more detail and try to explain the discrepancies concerning NRF2 in vitiligo (page 4).

4. In Section 4.6. Future directions, the authors emphasized the importance of the development of therapies that target the IFN-γ-CXCL10-CXC receptor type 3 (CXCR3) axis highlighting JAK inhibitors as successful treatment options for vitiligo. They could make a better link (if existing) with the NRF2 targeted therapy.

5. Some of the examples of recent research papers that could add to this manuscript:

Kang P, Chen J, Zhang W, Guo N, Yi X, Cui T, Chen J, Yang Y, Wang Y, Du P, Ye Z, Li B, Li C, Li S. Oxeiptosis: a novel pathway of melanocytes death in response to oxidative stress in vitiligo. Cell Death Discov. 2022 Feb 17;8(1):70. doi: 10.1038/s41420-022-00863-3.

Shi Q, Zhang W, Guo S, Jian Z, Li S, Li K, Ge R, Dai W, Wang G, Gao T, Li C. Oxidative stress-induced overexpression of miR-25: the mechanism underlying the degeneration of melanocytes in vitiligo. Cell Death Differ. 2016 Mar;23(3):496-508. doi: 10.1038/cdd.2015.117.

Mou K, Liu W, Miao Y, Cao F, Li P. HMGB1 deficiency reduces H2 O2 -induced oxidative damage in human melanocytes via the Nrf2 pathway. J Cell Mol Med. 2018 Dec;22(12):6148-6156. doi: 10.1111/jcmm.13895.

Romano-Lozano V, Cruz-Avelar A, Peralta Pedrero ML. Nuclear Factor Erythroid 2-Related Factor 2 in Vitiligo. Actas Dermosifiliogr. 2022 Jul-Aug;113(7):705-711. English, Spanish. doi: 10.1016/j.ad.2022.02.025.

Shin JM, Kim MY, Sohn KC, Jung SY, Lee HE, Lim JW, Kim S, Lee YH, Im M, Seo YJ, Kim CD, Lee JH, Lee Y, Yoon TJ. Nrf2 negatively regulates melanogenesis by modulating PI3K/Akt signaling. PLoS One. 2014 Apr 24;9(4):e96035. doi: 10.1371/journal.pone.0096035.

Author Response

We thank the Reviewers for taking time to review our manuscript and giving us valuable suggestions. We have provided a detailed point-by-point response to their comments, which we found to be very helpful. We believe that our revised manuscript has been greatly improved by the suggestions, and hope that it meets with their approval.

  1. Section 2. “The KEAP1-NRF2 System for Maintaining Redox Homeostasis” could be more clearly written. The authors state: “The cytoplasmic protein KEAP1 senses oxidative insults, whereas the transcription factor NRF2 exerts counter responses [4]. In the steady state, NRF2 is polyubiquitinated and degraded by proteasomes. Upon sensing electrophiles or ROS, the reactive cysteine residues of KEAP1 are covalently modified, resulting in NRF2 stabilization….”

What do the authors mean by KEAP1 senses oxidative insults while NRF2 exerts counter-responses? In addition, when describing the NRF2/KEAP1 system in a steady state, how are covalent modifications of KEAP1 cysteine residues stabilizing NRF2? Which NRF2 activates NRF2 target genes?

Response: Thank you for flagging this issue. In the physiological state, NRF2 is ubiquitinated by the KEAP1-CUL3 ubiquitin E3 ligase, which marks NRF2 for rapid degradation by the proteasome. However, when cells are exposed to electrophiles or ROS, the cysteine residues of KEAP1 are covalently modified and the KEAP1-CUL3 ubiquitin E3 ligase activity declines, resulting in NRF2 stabilization. Stabilized and accumulated NRF2 translocates to the nucleus and robustly activates a battery of cytoprotective genes. Therefore, KEAP1 and NRF2 are considered as “sensor” and “effector” respectively. We have carefully described the molecular mechanism of the KEAP1-NRF2 system (please see section 2 in our revised manuscript).

  1. Page 3, lines 139-140, sentence “ROS are released from melanocytes in response to stress and can lead to impaired expression or activity of the antioxidant system.” Unclear sentence. How is ROS released from melanocytes in response to stress?

Response: We thank you for pointing it out. Although the precise mechanism of ROS production in melanocytes is unclear, mitochondria are considered a key source of ROS, thereby mediating melanocyte dysfunction. We have modified the description accordingly (please see section 4.2 in our revised manuscript).

  1. The authors should discuss in more detail and try to explain the discrepancies concerning NRF2 in vitiligo (page 4).

Response: We appreciate your valuable comment.  We have discussed more deeply the role of NRF2 in melanocyte biology and vitiligo pathogenesis (please see section 4.4 in our revised manuscript). Evidence indicates that NRF2 supports melanocyte survival, and thus aberrant antioxidative system can be central to vitiligo pathogenesis. Although it may be considered contradictory, the lesional epidermis of patients with vitiligo exhibited higher NRF2 and its downstream target mRNA expression levels compared to non-lesional epidermis. We presume that this observation was owing to higher oxidative stress in the lesional epidermis. Supporting this view, the electrophile treatment induced lower phase II detoxification genes in the lesional skin.  Besides, the electrophile treatment resulted in distinct susceptibility to apoptosis between melanocytes and keratinocytes.

Collectively, we assume that melanocyte-keratinocyte interaction is indispensable for functional antioxidant response in the epidermis.

  1. In Section 4.6. Future directions, the authors emphasized the importance of the development of therapies that target the IFN-γ-CXCL10-CXC receptor type 3 (CXCR3) axis highlighting JAK inhibitors as successful treatment options for vitiligo. They could make a better link (if existing) with the NRF2 targeted therapy.

Response: Thank you for your comment. Cold atmospheric plasma, one of the NRF2-targeted therapies for vitiligo, decreased CD8+ T cell infiltration and IFN-γ and CXCL10 release in mouse vitiligo lesions, which may associate NRF2 and immune modulation (please see Table 1).

  1. Some of the examples of recent research papers that could add to this manuscript:

Kang P, Chen J, Zhang W, Guo N, Yi X, Cui T, Chen J, Yang Y, Wang Y, Du P, Ye Z, Li B, Li C, Li S. Oxeiptosis: a novel pathway of melanocytes death in response to oxidative stress in vitiligo. Cell Death Discov. 2022 Feb 17;8(1):70. doi: 10.1038/s41420-022-00863-3.

Shi Q, Zhang W, Guo S, Jian Z, Li S, Li K, Ge R, Dai W, Wang G, Gao T, Li C. Oxidative stress-induced overexpression of miR-25: the mechanism underlying the degeneration of melanocytes in vitiligo. Cell Death Differ. 2016 Mar;23(3):496-508. doi: 10.1038/cdd.2015.117.

Mou K, Liu W, Miao Y, Cao F, Li P. HMGB1 deficiency reduces H2 O2 -induced oxidative damage in human melanocytes via the Nrf2 pathway. J Cell Mol Med. 2018 Dec;22(12):6148-6156. doi: 10.1111/jcmm.13895.

Romano-Lozano V, Cruz-Avelar A, Peralta Pedrero ML. Nuclear Factor Erythroid 2-Related Factor 2 in Vitiligo. Actas Dermosifiliogr. 2022 Jul-Aug;113(7):705-711. English, Spanish. doi: 10.1016/j.ad.2022.02.025.

Shin JM, Kim MY, Sohn KC, Jung SY, Lee HE, Lim JW, Kim S, Lee YH, Im M, Seo YJ, Kim CD, Lee JH, Lee Y, Yoon TJ. Nrf2 negatively regulates melanogenesis by modulating PI3K/Akt signaling. PLoS One. 2014 Apr 24;9(4):e96035. doi: 10.1371/journal.pone.0096035.

Response: We appreciate this valuable suggestion. We have incorporated those recent reports regarding oxidative stress/NRF2 and melanocyte biology/vitiligo pathogenesis to our manuscript (please see section 4.2. and 4.4. in our revised manuscript).

Reviewer 2 Report

The authors have given an account of the major function of NRF2 in melanocyte biology, vitiligo pathogenesis, and therapeutic approaches. The review written by the authors Ogawa and Ishitsuka is well written However there are several major concerns that must be addressed and shown below: 

1. The abstract of the manuscript is a bit confusing for the readers. Authors should clearly mention what is the basic purpose of the review. Are the authors talking about NRF2 as a defense against vitiligo, pathogenic mechanisms, or both? It should be clearly mentioned in the abstract. Briefly but clearly in the abstract.

2. The introduction is very basic and minimal. It is hardly conveying any special information. The information about the NRF2-KEAP-1 association is already so much discussed topic. The introduction should be deeper and more meaningful. 

3. Authors have nowhere mentioned the prevalence of vitiligo and give some statistics on the last 10 years of the condition on a global level.

4. What is ARE? It should be explained in page no 1 line no 37. 

5. What is α-MSH? Explain on page no 2 line no 95.

6. Make the conclusion more simple and explanatory for better readability.

7. There are a lot of abbreviations in the manuscript and their full form is missing. The authors should rectify the error.

8. Kindly explain the novelty of the review in the introduction part of the manuscript which will give a better insight into reading the manuscript.

9. Some more figures should be added to the manuscript as only one figure makes this review very basic. Authors should enhance the quality of the manuscript as the information given is not up to the mark and is very basic.

10. Authors have mostly used a lot of old references. Kindly update your reference list.

Author Response

We thank the Reviewers for taking time to review our manuscript and giving us valuable suggestions. We have provided a detailed point-by-point response to their comments, which we found to be very helpful. We believe that our revised manuscript has been greatly improved by the suggestions, and hope that it meets with their approval.

  1. The abstract of the manuscript is a bit confusing for the readers. Authors should clearly mention what is the basic purpose of the review. Are the authors talking about NRF2 as a defense against vitiligo, pathogenic mechanisms, or both? It should be clearly mentioned in the abstract. Briefly but clearly in the abstract.

Response: Thank you for flagging this important issue. We meant that NRF2 supports melanocyte survival against oxidative stress, and thus aberrant NRF2-mediated antioxidant system is involved in vitiligo pathogenesis. We have modified the description accordingly in the abstract.

  1. The introduction is very basic and minimal. It is hardly conveying any special information. The information about the NRF2-KEAP-1 association is already so much discussed topic. The introduction should be deeper and more meaningful.

Response: We thank you for giving us this notion. We have additionally described overview of oxidative stress/ROS in the introduction section and modified a logical flow.

  1. Authors have nowhere mentioned the prevalence of vitiligo and give some statistics on the last 10 years of the condition on a global level.

Response: We already described the prevalence of vitiligo as follows: “…that affects 0.5% to 2% of the world’s population without a clear preference for race or sex.” We have additionally described the US population-based prevalence of vitiligo, which was conducted a few years before (JAMA Dermatology 2022; 158: 43-50) (please see page 4 line 161-163).

  1. What is ARE? It should be explained in page no 1 line no 37.

Response: We have described ARE as antioxidant response element or electrophile responsive element (EpRE), to which nuclear translocated NRF2 binds (please see page 2 line 59 in our revised manuscript).

  1. What is α-MSH? Explain on page no 2 line no 95.

Response: We have referred to α-MSH as α-melanocyte-stimulating hormone, which mediates melanocyte-keratinocyte crosstalk (please see page 2 line 80 in our revised manuscript).

  1. Make the conclusion more simple and explanatory for better readability.

Response: Thank you for your suggestion. We have incorporated conclusions into the future directions section and presented a concept that keratinocytes may imprint epidermal melanocyte biology.

  1. There are a lot of abbreviations in the manuscript and their full form is missing. The authors should rectify the error.

Response: We already incorporated abbreviation list (please see page 12-13 in our revised manuscript).

  1. Kindly explain the novelty of the review in the introduction part of the manuscript which will give a better insight into reading the manuscript.

Response: We appreciate your meaningful comment. We have modified the aim description in the introduction section to clearly convey significance of this review to the readers.

  1. Some more figures should be added to the manuscript as only one figure makes this review very basic. Authors should enhance the quality of the manuscript as the information given is not up to the mark and is very basic.

Response: We appreciate your valuable suggestion. We have additionally created a figure that delineate melanogenesis process and resultant ROS production (please see Figure 1 in our revised manuscript).

  1. Authors have mostly used a lot of old references. Kindly update your reference list.

Response: Thank you for your valuable comment. We have incorporated some recent reports regarding oxidative stress/NRF2 and melanocyte biology/vitiligo pathogenesis to our revised manuscript.

Round 2

Reviewer 1 Report

The authors made significant changes to the previously submitted review paper. Some of the questions that are now puzzling are:

1. Why have they changed their view/conclusion from emphasizing the importance of NRF2 and thus suggesting it as an attractive therapeutic target to, as they say, " In this review, we initially aimed to examine the role of the KEAP1-NRF2 system in melanocyte biology/vitiligo pathogenesis. It has turned out, however, that this role [5] appears too far-reaching to be a disease-specific pathway.” Nevertheless, they conclude, " We and others have characterized the roles of the KEAP1-NRF2 system in skin diseases involving aberrations in inflammation/keratinization (reviewed in [9]). The aggregated evidence underscores the prominent roles of the KEAP1-NRF2 system in epidermal biology”.

Are the authors suggesting that the NRF2/KEAP1 system is important in epidermal biology and vitiligo, but not so specific as to be a therapeutic target?

2. Subsection "4.6. Future directions and conclusions" is part of section 4. Vitiligo, although it should be a separate section. In this subsection, apart from a few sentences about the NRF2/KEAP1 system (some already mentioned in the previous comment), they focus on SE as a pigmentary "unit", "the niche", the importance of DSG1, and loricrin. Their claims have merit, but they are not really linked to the NRF2 system. Also, the authors went a little overboard with the self-citations in this part.

9. Ishitsuka, Y.; Ogawa, T.; Roop, D. The KEAP1/NRF2 Signaling Pathway in Keratinization. Antioxidants (Basel) 2020, 9, 1033

doi:10.3390/antiox9080751.

140. Ishitsuka, Y.; Roop, D.R. Loricrin at the Boundary between Inside and Outside. Biomolecules 2022, 12, doi:10.3390/biom12050673.

142. Ishitsuka, Y.; Roop, D.R. Loricrin: Past, Present, and Future. Int J Mol Sci 2020, 21, doi:10.3390/ijms21072271.

143. Ishitsuka, Y.; Roop, D.R. Loricrin Confers Photoprotective Function against UVB in Corneocytes. J Invest Dermatol 2018, 138, 2684-2687, doi:10.1016/j.jid.2018.06.164. 1383Biomolecules 2022, 12, x FOR PEER REVIEW 23 of 23

144. Ogawa, T.; Ishitsuka, Y.; Nakamura, Y.; Watanabe, R.; Okiyama, N.; Fujisawa, Y.; Fujimoto, M.; Roop, D.R.; Nomura, T. Loricrin Protects against Chemical Carcinogenesis. J Invest Dermatol 2021, doi:10.1016/j.jid.2021.12.015.

134. Wakabayashi, N.; Itoh, K.; Wakabayashi, J.; Motohashi, H.; Noda, S.; Takahashi, S.; Imakado, S.; Kotsuji, T.; Otsuka, F.; Roop, D.R.; et al. Keap1-null mutation leads to postnatal lethality due to constitutive Nrf2 activation. Nat Genet 2003, 35, 238-245, doi:10.1038/ng1248

Specific comment:

3. The authors state: “Melanins come in two flavors:…”

I would suggest other words instead of flavors, such as types or forms. And melanin, not melanins.

4. Paragraph “Augmented NRF2 signaling could replenish the cellular GSH pool (thus increasing the GSH/GSSG ratio) [5] and support melanocyte proliferation while hampering eumelanogenesis [34]. Accordingly, adenovirus-mediated NRF2 overexpression inhibits melanogenesis in NHEMs [6]. Conversely, knockdown of NRF2/its downstream antioxidants (NQO1 and PRDX6) [94] or sut mice pigmentary mutation in the Slc7a11 gene reduces cultured melanocyte viability [34].”

Reference 34. Chintala, S.; Li, W.; Lamoreux, M.L.; Ito, S.; Wakamatsu, K.; Sviderskaya, E.V.; Bennett, D.C.; Park, Y.M.; Gahl, W.A.; Huizing, M.; et al. Slc7a11 gene controls production of pheomelanin pigment and proliferation of cultured cells. Proc Natl Acad Sci U S A 2005, 102, 10964-10969, doi:10.1073/pnas.0502856102. shows “that Slc7a11 is a major genetic regulator of pheomelanin pigment in hair and melanocytes, with minimal or no effects on eumelanin. Furthermore, transport of cystine by xCT is critical for normal proliferation, glutathione production, and protection from oxidative stress in cultured cells.”

Therefore, the authors should cite it more adequately.

Author Response

The authors made significant changes to the previously submitted review paper. Some of the questions that are now puzzling are:

1. Why have they changed their view/conclusion from emphasizing the importance of NRF2 and thus suggesting it as an attractive therapeutic target to, as they say, " In this review, we initially aimed to examine the role of the KEAP1-NRF2 system in melanocyte biology/vitiligo pathogenesis. It has turned out, however, that this role [5] appears too far-reaching to be a disease-specific pathway.” Nevertheless, they conclude, " We and others have characterized the roles of the KEAP1-NRF2 system in skin diseases involving aberrations in inflammation/keratinization (reviewed in [9]). The aggregated evidence underscores the prominent roles of the KEAP1-NRF2 system in epidermal biology”.

Are the authors suggesting that the NRF2/KEAP1 system is important in epidermal biology and vitiligo, but not so specific as to be a therapeutic target?

Thank you very much for pointing out this critical issue. We agree to the Reviewer’s view and revised the manuscript according to the valuable suggestions. We also summed up section 4.6. with a conclusion statement (L335).

2. Subsection "4.6. Future directions and conclusions" is part of section 4. Vitiligo, although it should be a separate section. In this subsection, apart from a few sentences about the NRF2/KEAP1 system (some already mentioned in the previous comment), they focus on SE as a pigmentary "unit", "the niche", the importance of DSG1, and loricrin. Their claims have merit, but they are not really linked to the NRF2 system. Also, the authors went a little overboard with the self-citations in this part.

We sincerely appreciate these critical but constructive comments.We separated the latter section of 4.6. because the content is not directly related to the NRF2’s effects on melanocytes or vitiligo.  As the Reviewer pointed out, this might have been puzzling at first glance. Nonetheless, our hypothetical claim may address the long-standing question of why only the interfollicular epidermis experience tanning among the stratified squamous epithelium (L354). The notion that the epidermal niche controls melanocytes’ biological behavior may also address the distinctive differences in the pathology/biology of cutaneous vs. acral/mucosal melanomas. However, this is definitely out of the scope of this manuscript. We do agree that the part was overloaded with self-citations and addressed the issue accordingly.

Specific comment:

3. The authors state: “Melanins come in two flavors:…”

I would suggest other words instead of flavors, such as types or forms. And melanin, not melanins.

We revised the description according to the comment.

4. Paragraph “Augmented NRF2 signaling could replenish the cellular GSH pool (thus increasing the GSH/GSSG ratio) [5] and support melanocyte proliferation while hampering eumelanogenesis [34]. Accordingly, adenovirus-mediated NRF2 overexpression inhibits melanogenesis in NHEMs [6]. Conversely, knockdown of NRF2/its downstream antioxidants (NQO1 and PRDX6) [94] or sut mice pigmentary mutation in the Slc7a11 gene reduces cultured melanocyte viability [34].”

Reference 34. Chintala, S.; Li, W.; Lamoreux, M.L.; Ito, S.; Wakamatsu, K.; Sviderskaya, E.V.; Bennett, D.C.; Park, Y.M.; Gahl, W.A.; Huizing, M.; et al. Slc7a11 gene controls production of pheomelanin pigment and proliferation of cultured cells. Proc Natl Acad Sci U S A 2005, 102, 10964-10969, doi:10.1073/pnas.0502856102. shows “that Slc7a11 is a major genetic regulator of pheomelanin pigment in hair and melanocytes, with minimal or no effects on eumelanin. Furthermore, transport of cystine by xCT is critical for normal proliferation, glutathione production, and protection from oxidative stress in cultured cells.”

Therefore, the authors should cite it more adequately.

Thank you for reviewing carefully, and we revised the description accordingly.